# Hydrological Impacts of Large Fires and Future Climate: Modeling Approach Supported by Satellite Data

**Claudia Carvalho-Santos [1,2,]*, Bruno Marcos [1,3], João Pedro Nunes [4], Adrián Regos [1,5], Elisa Palazzi [6], Silvia Terzago [6], António T. Monteiro [1,7] and João Pradinho Honrado [1,3]**

[1] CIBIO-InBIO—Centro de Investigação em Biodiversidade e Recursos Genéticos, Laboratório Associado, Universidade do Porto, Campus Agrário Vairão, 4485-661 Vila do Conde, Portugal; bruno.marcos@cibio.up.pt (B.M.); adrian.regos@cibio.up.pt (A.R.); monteiroantonio@edu.ulisboa.pt (A.T.M.); jhonrado@fc.up.pt (J.P.H.)

[2] CBMA—Centre of Molecular and Environmental Biology, University of Minho, 4710–057 Braga, Portugal

[3] Faculdade de Ciências, Universidade do Porto, Rua do Campo Alegre, FC4-Biologia, 4169-007 Porto, Portugal

[4] CE3C—Centre for Ecology, Evolution and Environmental Changes, Faculdade de Ciências, Universidade de Lisboa, Campo Grande 016, 1749-016 Lisboa, Portugal; jpcnunes@fc.ul.pt

[5] Departamento de Zooloxía, Xenética e Antropoloxía Física, Universidade de Santiago de Compostela, 15782 Santiago de Compostela, Spain

[6] Institute of Atmospheric Sciences and Climate, National Research Council (ISAC-CNR), Corso Fiume 4, 10133 Torino, Italy; e.palazzi@isac.cnr.it (E.P.); s.terzago@isac.cnr.it (S.T.)

[7] Institute of Geography and Spatial Planning- MOPT unit, University of Lisbon, Rua Branca Edmée Marques, 1600-276 Lisbon, Portugal

* Correspondence: c.carvalho.santos@cibio.up.pt or claudiasantos.malta@gmail.com; Tel.: +351-252-660-411

**Abstract:** Fires have significant impacts on soil erosion and water supply that may be exacerbated by future climate. The aims of this study were: To simulate the effects of a large fire event in the SWAT (Soil and Water Assessment Tool) hydrological model previously calibrated to a medium-sized watershed in Portugal; and to predict the hydrological impacts of large fires and future climate on water supply and soil erosion. For this, post-fire recovery was parametrized in SWAT based on satellite information, namely, the fraction of vegetation cover (FVC) calculated from the normalized difference vegetation index (NDVI). The impact of future climate was based on four regional climate models under the stabilization (RCP 4.5) and high emission (RCP 8.5) scenarios, focusing on mid-century projections (2020–2049) compared to a historical period (1970–1999). Future large fire events (>3000 ha) were predicted from a multiple linear regression model, which uses the daily severity rating (DSR) fire weather index, precipitation anomaly, and burnt area in the previous three years; and subsequently simulated in SWAT under each climate model/scenario. Results suggest that time series of satellite indices are useful to inform SWAT about vegetation growth and post-fire recovery processes. Different land cover types require different time periods for returning to the pre-fire fraction of vegetation cover, ranging from 3 years for pines, eucalypts, and shrubs, to 6 years for sparsely vegetated low scrub. Future climate conditions are expected to include an increase in temperatures and a decrease in precipitation with marked uneven seasonal distribution, and this will likely trigger the growth of burnt area and an increased frequency of large fires, even considering differences across climate models. The future seasonal pattern of precipitation will have a strong influence on river discharge, with less water in the river during spring, summer, and autumn, but more discharge in winter, the latter being exacerbated under the large fire scenario. Overall, the decrease in water supply is more influenced by climate change, whereas soil erosion increase is more dependent on fire, although with a slight increase under climate change. These results emphasize the need for adaptation measures that target the combined hydrological consequences of future climate, fires, and post-fire vegetation dynamics.

**Keywords:** future climate; fire; hydrological impacts; post-fire recovery; satellite data; SWAT model

## 1. Introduction

Water supply (in terms of quantity, seasonality, and quality) and the control of soil erosion in watersheds worldwide are threatened by various pressures, such as fires and climate change, that affect the functioning of ecosystems [1,2]. Fires reduce soil infiltration due to the formation of a topsoil hydrophobicity layer, and they promote surface runoff and soil erosion with consequences for water quality [3,4]. These negative effects of fires may be exacerbated by future climate change [5].

Although fire plays a role in the dynamics of many ecosystems, influencing biodiversity and landscape heterogeneity [6], they are considered a natural hazard, becoming more frequent, intense, and large, associated to more economic and social impacts, often with human fatalities [7]. Even if wildfires are a common feature in regions under the Mediterranean type of climate, the determinants of fire regimes are rapidly modified by changes in the landscape, climate, and socio-economic factors [4,5,7]. Weather conditions favorable for fires, in combination with ignition sources and fuel availability, are projected to increase in the future [8]. However, there is substantial uncertainty regarding vegetation–fire feedbacks and how they will influence future shifts in fire regimes [5,9].

The hydrological impacts of management practices (e.g., in agriculture), land cover, and climate change have long been assessed using hydrological modeling, and in particular, using SWAT (Soil and Water Assessment Tool), a widely used hydrological model developed by the United States Department of Agricultural Research (USDA) in the early 1990s [10]. Although post-fire runoff and soil erosion have long been evaluated using other hydrological/soil erosion models [11–14], only recently they have started being modeled in SWAT. In some cases through a simulation of different scenarios of land cover thus approaching post-fire vegetation conditions in a static way [15], in other cases by performing a statistical analysis rating infiltration and flow as the proportion of area burned [16], or modifying key model parameters in post-fire conditions [17], coupled with a land-use update model [18]. All of these approaches focus on evaluating the water balance components, while excluding soil erosion. In general, these simulations predict a reduction of water storage due to decrease in infiltration, and an increase in peak flows resulting in the high probability of floods occurrence after fire. Recently, SWAT was applied to evaluate both the water balance and soil erosion, modifying key parameters and validating for observed streamflow and erosion in burnt hillslopes in a watershed in central Portugal; the study concluded that fire-prone forests might not provide the anticipated soil protection and water quality regulation services [19].

However, in all of these studies, climate change and fire scenarios for the future were never addressed. Moreover, previous studies advocate the need to include vegetation dynamics through process-based models to improve models of fire probability under climate change [20,21]. In this regard, satellite data can play a crucial role to inform ecosystem models, either as input data, to validate/calibrate model results, or as a tool to sequentially update variables, reducing model uncertainty [22]. Therefore, here we combined SWAT and satellite data to assess the hydrological and erosional impacts of fires, and to simulate those effects combined with future climate change. The study took advantage of a previous application of SWAT in the Vez watershed (Portugal) and updated the model with a calibration for large fire events and post-fire recovery, which was informed by satellite data. The main novelty resides in simulating the consequences of both future climate and fire regime on water quantity and seasonality and on soil erosion, in a dynamic way, particularly considering post-fire vegetation regrowth.

## 2. Materials and Methods

### 2.1. Study-Area

The Vez is a subsidiary of river Lima, and drains a medium-sized watershed (260 km$^2$) in northwest Portugal. This mountainous watershed, mainly set on granite bedrock, has a complex topography with altitudes ranging from 30 to 1400 m and slopes above 25% in half of the watershed. It receives high levels of precipitation all year, with the exception of two drier months in summer (July and August). Annual precipitation is about 1500 mm and average temperature is 13 °C. The main land cover in the watershed is shrubland (tall and dwarf), followed by forests (of oaks, pines, and eucalypts) and agriculture in the valleys rotating grass in the winter and corn in summer (Table 1).

**Table 1.** Land cover classes in the Vez watershed when a large fire occurs. Burnt in 2006, two classes were assigned for each land cover, to respect the geometry, e.g., FIMI and MIFI, before and after the fire, respectively. FIMI has the same vegetation parameters as MIGS; MIFI has the parameters for burnt shrub (Table 2).

| Condition | SWAT Class | % Land Cover |
|---|---|---|
| **Unburnt** **80%** | MIGS—shrub | 33 |
| | BSVG—low shrub | 7.5 |
| | CARV—oak | 7.5 |
| | EUCL—eucalypts | 1 |
| | PINU—pine | 7 |
| | CORN—agriculture (maize + pasture) | 19 |
| | URBN—urban low residential | 5 |
| **Burnt in 2006** **20%** | FIMI—shrub/MIFI—burnt shrub | 12 |
| | FIBS—low shrub/BSFI—burnt low shrub | 5.6 |
| | FICA—oak/CAFI—burnt oak | 1.4 |
| | FIEU—eucalypts/EUFI—burnt eucalypts | 0.3 |
| | FIPI—pine/PIFI—burnt pine | 0.7 |

This watershed is frequently affected by fires, especially in summer. Particularly, a large fire event occurred from the 8–15 August 2006, burning 23% of the watershed (i.e., about 8000 ha). The event was characterized by three fires in different places (Figure 1a. Part of the affected area was inside of the Peneda-Gerês National Park, in particular, inside the special zone for protection of old-growth oak (*Quercus robur*) forests, an important habitat for several endangered fauna and flora species. Because the landscape is highly prone to fires in the dry periods, namely in areas with flammable vegetation types (shrubland, pines, and eucalypts), abandoned land, and steep slopes, there is high fire recurrence, with some areas experiencing 4–6 fires within 16 years (Figure 1b). The frequency of years with total burnt area above 6000 hayr$^{-1}$ is increasing in the last decade, as well as years with the mean area burnt per fire event above 500 ha, set as the value to be considered as a large fire (Figure 1c).

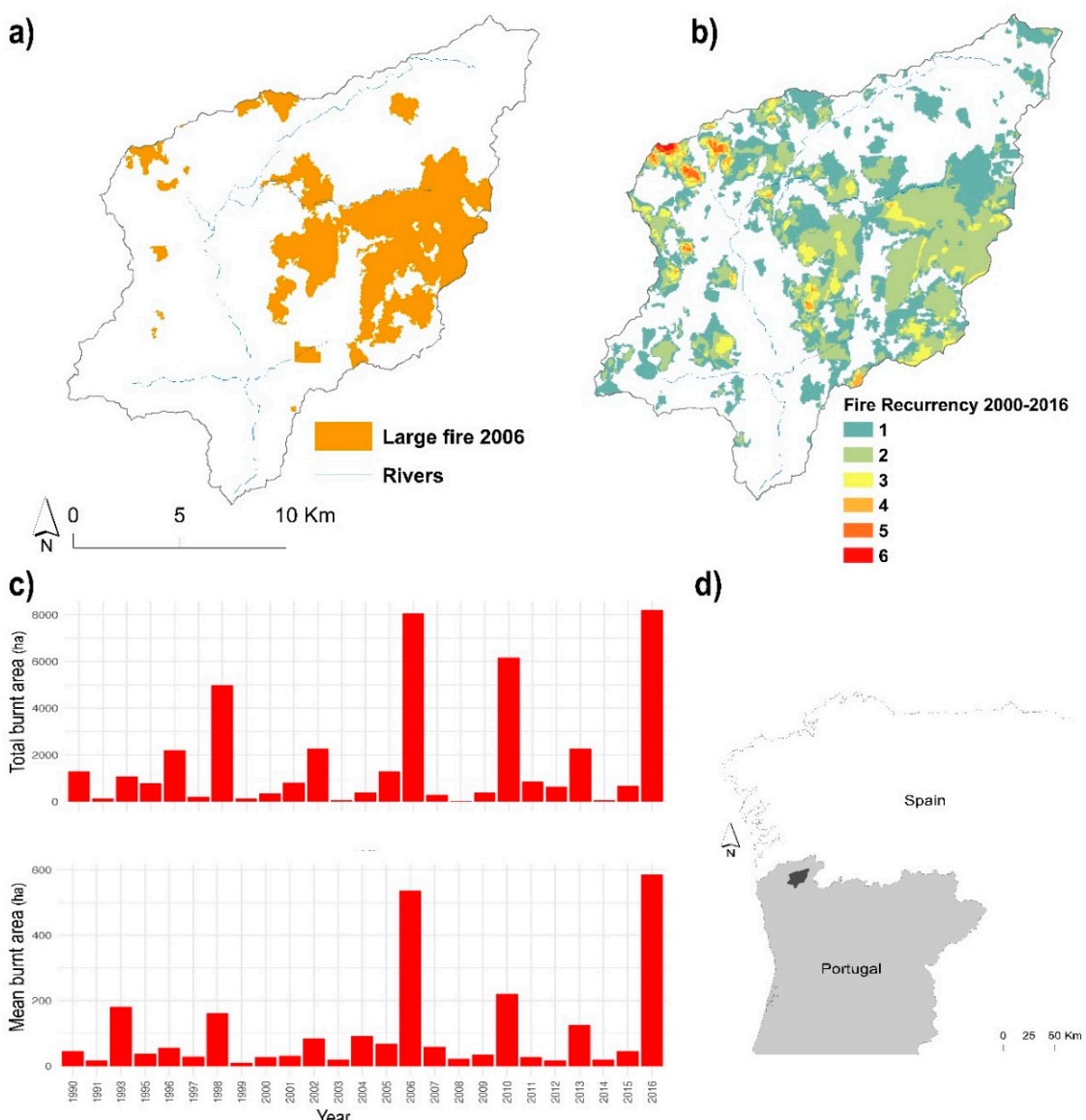

**Figure 1.** Fires in the Vez watershed. (**a**) Burnt area in the large fire event that occurred in August 2006 in the Vez watershed; (**b**) fire recurrence (years) in 16 years (2000–2016); (**c**) total and mean per fire event burnt area in the watershed (1990–2016); (**d**) the Vez watershed location in northwest Portugal.

## 2.2. Satellite Data to Support SWAT Parametrization

The utility of satellite-derived data to support hydrological modeling, in particular, SWAT, is growing in the last years, mainly for watersheds with in-situ data scarcity. Examples range from the calibration and validation of evapotranspiration process [23], to using rainfall satellite products forcing the SWAT model in places where in-situ data is absent [24]. Here, satellite data were used to inform the vegetation growth model in SWAT: First to calibrate leaf area index (LAI) through vegetation parameters, and afterwards to calibrate post-fire recovery time steps by land cover type.

To calibrate vegetation parameters related to LAI, a time series from the LAI MODIS product (MOD15A2, collection 5, 8-day, 1 km) was used to extract LAI average values for different land cover types (assuming ≥75% cover as a pure land cover pixel) for the period 2003–2008, the same period of SWAT calibration in the watershed. This information was used to manually fine-tune vegetation parameters in SWAT, such as the minimum leaf area index (ALAI_MIN) (Table 2), to approximate SWAT LAI values and patterns with the ones observed from the satellite.

**Table 2.** SWAT model parameters for vegetation unburnt and burnt.

| Parameter | Description | Vegetation Unburnt (Burnt) | | | | |
|---|---|---|---|---|---|---|
| | | Pine FIPI (PIFI) | Eucalypts FIEU (EUFI) | Oak FICA (CAFI) | Shrub FIMI (MIFI) | Low Shrub FIBS (BSFI) |
| BLAI | Maximum potential leaf area index ($m^2/m^2$) | 4 (1) | 3.7 (1) | 6 (1) | 2 (0.5) | 5 (0.5) |
| ALAI_MIN | Minimum leaf area index for plant during dormant period ($m^2/m^2$) | 3.9 (1) | 3.4 (1) | 0.75 (1) | 1.8 (0) | 0 (0) |
| USLE_C | Minimum value of USLE C factor for water erosion (factor) | 0.001 (0.004) | 0.001 (0.008) | 0.001 (0.002) | 0.001 (0.004) | 0.005 (0.008) |
| OV_N (.hru) | Curve number for moisture condition | 0.8 (0.1) | 0.4 (0.1) | 0.8 (0.2) | 0.8 (0.1) | 0.17 (0.1) |
| PHU_PLT | Heat units to maturity | 3500 (1500) | 3500 (1500) | 3500 (1500) | 2500 (1000) | 1500 (500) |

Secondly, time-series of the fraction of vegetation cover (FVC) was estimated from the normalized difference vegetation index (NDVI), extracted from MODIS/Terra image time-series (MODIS product MOD13Q1, 16-day, 250 m), by applying the formula [25]:

$$FVC = \frac{NDVI - NDVI_{min}}{NDVI_{max} - NDVI_{min}} \tag{1}$$

where $NDVI_{min}$ and $NDVI_{max}$ correspond to NDVI values for bare soil (FVC = 0) and dense vegetation (FVC = 1), respectively. Using FVC facilitates interpretation, since it is not unitless, unlike NDVI. This way, an estimate of FVC for each $250 \times 250$ m square, for every 16 days, was obtained. These values allowed us to monitor post-fire recovery in the study area identified as burnt in 2006 in the MODIS burnt area product MCD64A1 (collection 6, monthly, 1 km). By grouping these temporal profiles by land cover class and calculating the median values using only pixels with a percentage area of that class equal or greater than 60%, we were able to extract median profiles of post-fire recovery for each land cover class.

Additionally, we estimated the time when vegetation started to regrow after fire, by identifying the first local minimum after fire (i.e., negative-to-positive inflexion point) in the long-term trend, obtained by using a seasonal–trend procedure based on the LOESS smoother (STL) time-series decomposition [26].

### 2.3. SWAT Hydrological Model Setup for Fire and Post-Fire Recovery

SWAT (Soil and Water Assessment Tool) is a widely used hydrological model to evaluate a broad range of environmental problems, such as the impact of agricultural practices, climate, and land cover changes on water resources [10]. It was successfully applied to the Vez watershed with climate and land cover change scenarios by Carvalho-Santos et al. [27], further developed with mapping and spatial analyses by Carvalho-Santos et al. [28]. However, in those previous studies, a major fire event that occurred in August 2006 was neglected during calibration procedures, mainly due to the lack of knowledge on post-fire consequences and how to address them with SWAT. Therefore, here we maintained all the steps for the SWAT application and calibration used in the previous study [27], further updated with a setup for fire (based on the one developed by Nunes et al. [19]) and with post-fire recovery (based on dynamic SWAT management operations and estimated from satellite data, as previously explained in Section 2.2). The software used was ArcSWAT version 2012.

The Vez watershed was divided in 10 sub-basins and 717 HRUs (hydrologic response units) with the same slope, land cover (Table 1), and soil type. The model was forced with daily climatic values (1999–2008 with 4 years of warming-up to setup initial conditions) for five local precipitation stations and one climatic station for maximum and minimum temperature, solar radiation, relative

humidity, and wind speed. Ten elevation bands were created to increase the spatial representation of precipitation and temperature, with a precipitation lapse rate of 1100 mm/km and a temperature lapse rate of −5 °C/km. The vegetation and soil parameters, as well as sensitive parameters for streamflow and sediment calibration, were the same as in Carvalho-Santos et al. [27]. For crops, it was assumed that 20% of the agricultural area was under permanent pasture and 80% under a rotation between corn in summer and pasture in winter. Management operations were established accordingly, starting corn growth in May, including fertilizer and auto-irrigation for harvesting in September, followed by winter pasture starting in October to be harvested in the following spring (April).

Calibration and validation were made against daily discharge and total suspended solids measured in one station close to the watershed outlet (2003–2008) [27]. A good agreement between model predictions and field observations related with discharge and sediments was obtained for calibration and validation exercises. Model performance was considered good according to a Nash–Sutcliffe efficiency (NSE) of 0.76 and a percentage of bias (PBIAS) of −15% for calibration of discharge. The calibration of sediments was considered adequate, given the limited observed values, but supported with values and parameters from the literature [27].

Fire was parameterized as follows: In the selected HRUs, previously clipped for the fire event in 2006 (Figure 1a), fires were simulated by settling a "burn" operation in SWAT on 10 August 2006, with BURN_FRLB (fraction of vegetation that burns) at 0.9. Since this operation is designed to simulate small fires in farming operations and therefore does not activate soil erosion in SWAT, we also performed a "harvesting and kill" operation on 15 August 2006, to remove vegetation from the ground and thus simulate more realistically the impact of wildfires on soil erosion in SWAT. In the spring of 2007, vegetation covers started to grow in burnt areas (Figure 2a), according to parameters described in Table 2, for burnt vegetation types. SWAT parameters for burnt vegetation/land cover were based on a previous publication in a comparable watershed in Central Portugal [19]. From these parameters, USLE_C (land cover factor for soil erosion) was settled based on Fernández and Vega [29] that compared modeling soil erosion values after fire with observed ones in the neighboring region of Galicia (Spain); values below 0.008 for USLE_C were associated to low-intensity fires. LAI was established as 1 in the following years after fire to perform the shrub domination in the first period of recovery. Satellite information (Figure 2a) was used to setup post-fire recovery in SWAT, and thus reproduce vegetation regrowth and, consequently, the fire hydrological consequences for the land covers affected by fire (oak, pine, eucalypt, tall shrub, and low shrub in the top of the mountains). This was done in a rotation scheme according to the number of years each land cover returns to the level of pre-fire cover, representing post-fire conditions for soil erosion and water supply quantity and quality.

## 2.4. Future Climate Scenarios

Future climate projections were obtained from four realizations of the SMHI-RCA4 regional climate model (RCM), driven by four global climate models providing boundary conditions for the regional simulations, included in the European Coordinated Regional Climate Downscaling Experiment (EURO-CORDEX, http://www.euro-cordex.net/), which is part of the wider CORDEX Initiative. The finest spatial resolution of these simulations, 0.11 degrees latitude–longitude, overtakes those achieved in previous modeling coordinated experiments using hydrostatic regional climate models, and thus was considered for this study. The four driving GCMs (General Circulation Models), namely, CNRM-CM5, EC-EARTH, IPSL-CM5A-MR, and MPI-ESM-LR, belong to the ensemble of models used as a basis for the latest IPCC (Intergovernmental Panel for Climate Change) assessment report (AR5, IPCC 2013) and are part of the Coupled Model Intercomparison Project phase 5 (CMIP5). Future projections considered two representative concentration pathway (RCP) scenarios [30], namely, RCP 4.5 and RCP 8.5. RCP 4.5 is a stabilization scenario assuming that radiative forcing is stabilized at 4.5 W/m$^2$ by and after 2100, reaching a peak around the mid-21st century [31], while RCP 8.5, also referred to as "business as usual", hypothesizes a high pathway emission scenario characterized by

increasing greenhouse gas emissions over time and corresponding radiative forcing reaching values greater than 8.5 W/m$^2$ by 2100 and continuously increasing.

For this study, we considered the following RCA4 model variables extracted over the region of interest: Daily maximum and minimum temperature, precipitation, surface downwelling shortwave radiation, and relative humidity. The temperature and precipitation data were bias-adjusted considering as a reference the observed climatology of E-OBS [32], available for the European domain at 0.25 degrees (~25 km) spatial resolution from 1970 up to 2005. Bias correction was performed pixel-by-pixel in the chosen domain and consisted of the adjustment of the long-term climatology using additive correction factors for the temperature and multiplicative correction factors for the precipitation [33,34]. The bias-adjusted temperature and precipitation data were further downscaled using orographic (lapse-rate) correction for the former and a stochastic downscaling method, the RainFARM [35,36], for the latter. Regional climate model data were tested in their ability to reproduce the conditions observed during the historical period 1970–1999 and employed to project future evolution and changes under the RCP 4.5 and RCP 8.5 scenarios in the next decades (2020–2049).

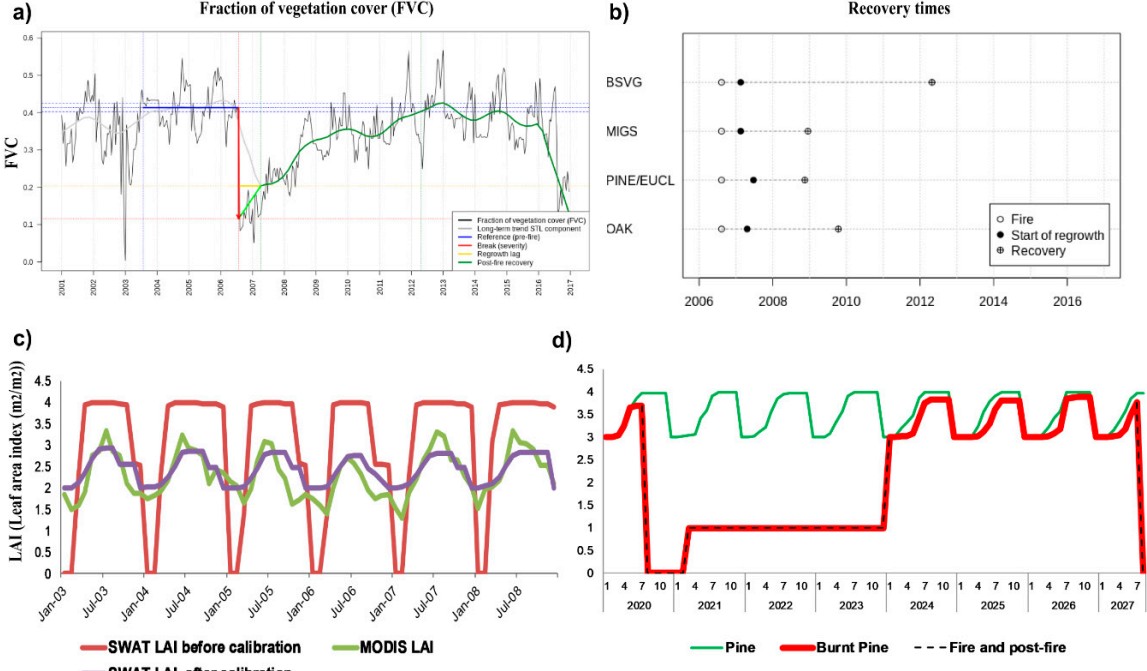

**Figure 2.** (**a**) Temporal profile of the fraction of vegetation cover (FVC) in a sample pixel, illustrating the procedure used for obtaining profiles of post-fire recovery and estimations of time to start vegetation regrowth after fire. (**b**) Average observed times for post-fire recovery by land cover type in the Vez watershed after the large fire event of August 2006 (BSVG—sparsely vegetated low shrub; MIGS—tall shrub). (**c**) Modeled leaf area index (LAI) of pine in SWAT, before and after vegetation parameter calibration, compared with MODIS LAI. (**d**) Modeled LAI of pine when a fire event is simulated in SWAT (the simulation shown is for 2020–2027, as the calibration period (2003–2008) was too short to reproduce LAI of post-fire).

## 2.5. Large Fires in the Future

Future projections of annual burnt area were made for the future climate scenarios following the empirical approach proposed by Sousa et al. [37]. Annual burnt area in the Vez watershed was modeled based on historical data for rainfall, fire weather index, and fire history, and compared to observed annual burnt area, between 1990–2005, freely available from the Portuguese Institute for Nature Conservation and Forests. Monthly rainfall anomalies (rainfall total—long-term average) were taken from E-OBS gridded dataset for Europe [32]. Fire weather was calculated based on the daily

severity rating (DSR) fire weather index [38], using climate data at midday for temperature, relative humidity, rainfall, and wind speed taken from the ERA-Interim reanalysis [39]. The analysis used the number of extreme fire weather days per month, calculated as the number of days with DSR above the 95th percentile from all values in spring, summer and autumn. Fire history was calculated as the accumulated burnt area in previous years.

A multiple linear regression model was developed by relating annual burnt area with (1) the rainfall anomalies from February to April and from July to August (representing the growing and summer dry seasons), (2) the extreme fire weather days between June and August (representing the main fire season), and (3) the total burnt area in the three previous years (representing fuel availability). Burnt area values calculated using this regression showed a good agreement with observations for the period 1990–2005 ($r^2$ = 0.69), even when using modeled burnt area values to represent burnt area in previous years, because before 1990 there were no data available ($r^2$ = 0.65). The parameters of the regression were similar to the ones calculated for the northwest Iberian Peninsula by Sousa et al. [37].

Future DSR values for 2020–2049 were calculated for all climate scenarios, and bias-corrected using an empirical quantile mapping approach [40]. These values were combined with rainfall projections (Section 2.3) to calculate annual burnt area for 2020–2049 for each climate scenario and model. The years with burnt area larger than 4000 ha, which is the 75th percentile of the historical burnt area in the watershed, were selected to run large fire events in SWAT. As some consecutive years with large fire events were predicted for the future (given the fire-prone meteorological conditions) and since only one map of burnt area in 2006 was used, we left at least 3 years between large fires to allow for vegetation to return to the pre-fire level of fraction cover (Figure 2a), and we searched instead for a year with a 3000 ha burnt area where the predicted fire could be simulated in SWAT (Table 3). This approach captured the favurable meteorological conditions for fire in the respective years, and also the post-fire meteorological conditions that strongly influence soil erosion. For the 4-year warming-up period (2016–2020), a fire was established in 2016 (Figure 1c). For the historical period (1970–1999), the same procedure was done, but without fire in the warming-up period. Both climate change only scenarios and climate change with large fire scenarios were performed in the same project in ArcSWAT to avoid some residual uncertainty.

## 3. Results and Discussion

### 3.1. Post-Fire Recovery in SWAT Based on Satellite Data

Satellite data were found to be very useful to parameterize vegetation growth processes in SWAT (Figure 2). MODIS LAI time-series helped to manually fine-tune vegetation parameters in SWAT to correctly reproduce growth cycle. In the example of pine (Figure 2c), SWAT LAI after calibration improved substantially, in particular to adjust minimum and maximum LAI to MODIS LAI values.

Regarding post-fire recovery, all land cover types started to regrow in the next spring after fire (March/April 2007) (Figure 2b). Different land cover types are associated with different average periods for returning to the level of pre-fire vegetation cover: 3 years for PINE/EUCL (mixed stands of pine and eucalypts) and MIGS (tall shrub); 4 years for OAK; and 6 years for BSVG (low shrub in the top of mountains). BSVG is the land cover type that takes more time to recover, probably due to the predominance of shallow soils in the top of the mountains, with depleted seed bank storage due to recurrent fires, and to the adverse climatic conditions such as strong winds and low temperatures that limit regrowth.

Areas dominated by eucalypts, pines, and tall shrubs take 3 years to return the same level of cover of pre-fire conditions, which may be related to the capacity of these species to quickly regenerate, much due to their higher capacity of surviving fire and resprouting. In fact, a study conducted in four plots of eucalypts after fire in central Portugal showed that this species is fire-resilient with a very high probability of surviving fire, depending on the respective fire severity, and the most common post-fire recovery type was basal resprouting [41]. In another study, NDVI time-series was used to model

post-fire vegetation recovery in Portugal, with results indicating that transitional woodland-shrub presents shorter recovery times, while coniferous tend to recover more slowly [42]. However, in our study, FVC was used instead of NDVI; the study area was smaller and did not contain the full range of vegetation types as in that study and the return to 100%, instead of 50%, of the pre-fire level was used as measure of post-fire recovery, which may show different results. Another source of uncertainty in our results is that the post-fire recovery curves were extracted from pixels with 75% (or higher whenever possible) of dominant land cover type. The small size and the large landscape heterogeneity of our study area prevents us to extract post-fire recovery curves from a significant number of "pure pixels" (i.e., 100% of dominant land cover type). In addition, our post-fire recovery curves were computed from pixels that may have burnt at different intensities, which can help explain the relatively slow post-fire recovery of fraction cover for some forest types.

The fraction of vegetation cover (FVC) by land cover type is a realistic approach to reproduce post-fire recovery in SWAT (Figure 2a). Thus, in the first years after fire until full recovery, LAI is low to simulate shrub dominance. Then, each land cover will eventually return to the pre-fire level of cover, according to the respective number of years. Overall, the SWAT model was able to reproduce post-fire recovery for each land cover type, with a marked difference in the evolution of LAI values through time when a fire was simulated (Figure 2b).

### 3.2. Climate and Large Fires in the Future

In general, less precipitation and a temperature increase are expected in the Vez watershed in 2020–2049 compared to the historical period (Figure 3a). However, climate models yield some variability in the range of temperature change and especially for precipitation. Although a reduction is expected, precipitation change ranges from about less 1% to about less 20% depending on the model and scenario. In one case, i.e., when the regional climate model is driven by the EC-EARTH global model in the RCP4.5 scenario, an increase in precipitation from about 8% is predicted.

In spite of the model ensemble spread in the annual precipitation change signal, the agreement among the models improves at the seasonal scale: All models under both scenarios, in fact, simulate more precipitation in winter and less in spring, summer, and autumn in the future, compared to the past. The same seasonal signal was also observed in the previous SWAT application to the Vez watershed, although using different climatic models and downscaling method [27].

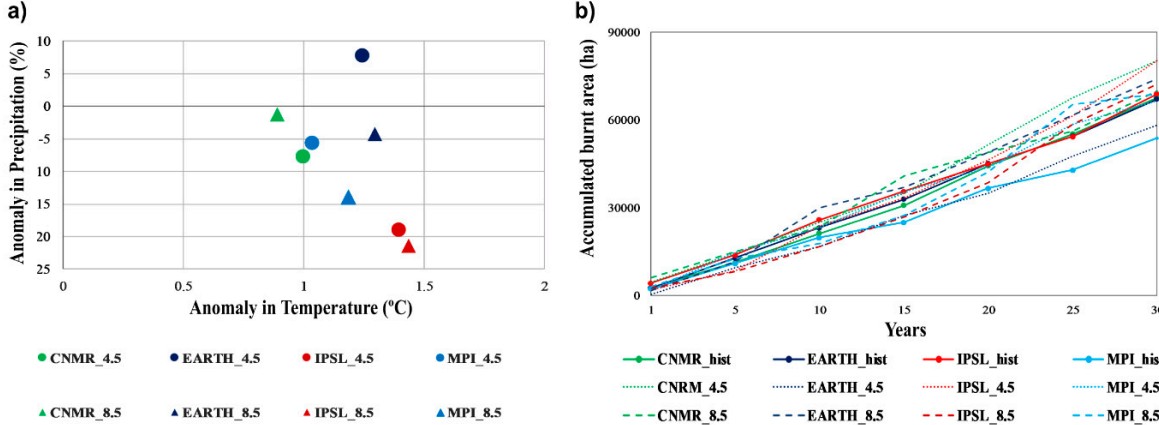

**Figure 3.** (**a**) Precipitation and temperature anomalies under future climate in the Vez watershed, by climate model, for scenarios RCP 4.5 and 8.5, for 2020–2049 compared to 1970–1999. (**b**) Accumulated burnt area for the future 30 years (2020–2049), by climate model and for scenarios RCP 4.5 and 8.5, compared to 30 years of historical simulation (1970–1999), derived by a multiple linear regression model.

Results from the multiple linear regression model for fire in the future show an increase in the total burnt area, with a maximum increase of about 28% found in MPI-ESM-LR-driven regional model simulations under the RCP 8.5 scenario (Table 3). A modeling study for the Mediterranean region forecasts a substantial increase in burnt areas in the future, especially for Portugal [43]. Accumulated burnt area is higher under the RCP 4.5 scenario than in RCP 8.5, especially for the CNRM-CM5- and IPSL-CM5A-MR-driven simulations (Figure 3b). Only the EC-EARTH-driven climate model, under the RCP 4.5 scenario, projects lower burnt area compared to historical conditions (Table 3), which is in line with the precipitation increase predicted by the same model (Figure 3a).

**Table 3.** Indicators for fire regime in Vez watershed projected in the future period 2020–2049 compared to the historical reference 1970–1999 (in brackets). The indicators result from a multiple linear regression model forced by the RCA4 regional climate model driven by four different GCMs (General Circulation Models) in RCP 4.5 and RCP 8.5 scenarios.

| | RCP 4.5 | | | | RCP 8.5 | | | |
|---|---|---|---|---|---|---|---|---|
| | CNRM-CM5 | EC-EARTH | IPSL-CM5A-MR | MPI-ESM-LR | CNRM-CM5 | EC-EARTH | IPSL-CM5A-MR | MPI-ESM-LR |
| Change in burnt area compared with each climate model historical (%) | 18.4 | −13.2 | 16.3 | 23.9 | 3.1 | 10.3 | 4.7 | 28.1 |
| Total burnt area (ha—30 years) | 80,267 | 58,292 | 80,342 | 66,701 | 69,889 | 74,060 | 72,294 | 68,974 |
| Maximum burnt area per year (ha) | 6 637 | 6 486 | 6 283 | 6 321 | 6 581 | 6 757 | 7 083 | 8 148 |
| Minimum burnt area per year (ha) | 0 | 6.5 | 31.8 | 0 | 44.2 | 0 | 0 | 0 |
| N° of years with burnt area | 29 | 30 | 30 | 29 | 30 | 28 | 29 | 27 |
| N° of years with low burnt area (<500 ha) (historical) | 3 (3) | 6 (0) | 4 (3) | 2 (1) | 5 | 4 | 2 | 5 |
| N° of years with large burnt area (>4000 ha) (historical) | 7 (3) | 2 (6) | 5 (3) | 2 (1) | 6 | 6 | 3 | 5 |
| N° of years with large burnt area (>3000 ha) (historical) | 11 (5) | 5 (3) | 12 (4) | 5 (3) | 7 | 9 | 9 | 10 |
| N° of years with fires simulated in SWAT (historical) | 5 (4) | 5 (4) | 7 (4) | 3 (3) | 4 | 5 | 4 | 4 |
| Total burnt area in SWAT (30 years) | 40,000 | 40,000 | 56,000 | 24,000 | 32,000 | 40,000 | 32,000 | 32,000 |

A higher frequency of large fires is expected in the future compared to the historical period, as shown by the calculated number of years with large burnt area (Table 3). However, previous research has found that past fires limit the growth of vegetation and, in turn, help suppress future large fires under climate change scenarios [44]. Although our regression model includes fire history as a predictor variable (being able to explain almost 70% of observed burnt area), simple correlative models hold important shortcomings. In the future, more sophisticated burnt area models using more variables and non-linear relationships [45,46] or more sophisticated approaches such as random forests and boosted regression trees [47] can be used to improve the robustness of burnt area predictions. The inclusion of landscape dynamics in the future burnt area model is beyond the scope of this research but is also worth exploring, considering the uncertainty in future climate vegetation feed-backs [9] and given the widespread land abandonment that has taken place in the study area over the last decades [48]. Nonetheless, we consider our model based on climatic conditions adequate to simulate future burnt areas to inform SWAT model. A previous study showed the best performance on modeling burnt area in the northwest region of Iberian Peninsula, where the Vez watershed is located, comparing with other regions, because of the strong relation between summer fires with climatic conditions in spring [49]. The same authors showed concerns for the future increase of the fire risk in this region due to the long-term reduction of the springtime precipitation, which is in line with our climate projections.

Overall, the increase in temperature and the marked seasonality of precipitation (more during winter and less in spring, summer, and autumn) projected for the future will definitely influence an increase of burnt areas in the Vez watershed, with expected increasing frequency of large fires events.

### 3.3. Hydrological Impacts of Future Large Fires under Climate Change

According to our simulations, climate models project an increase in temperatures and an intensification of the seasonality of precipitation in the Vez watershed for the future period 2020–2049. This will directly influence the river discharge, with moderate water reduction in summer and a more visible reduction in spring and autumn, especially under the RCP 4.5 climate scenario (Figure 4a). A small increase of discharge during winter is expected. These results from the ensemble of SWAT outputs differ across climate models, which shows the amplitude of variance in simulated discharge (Figure 4).

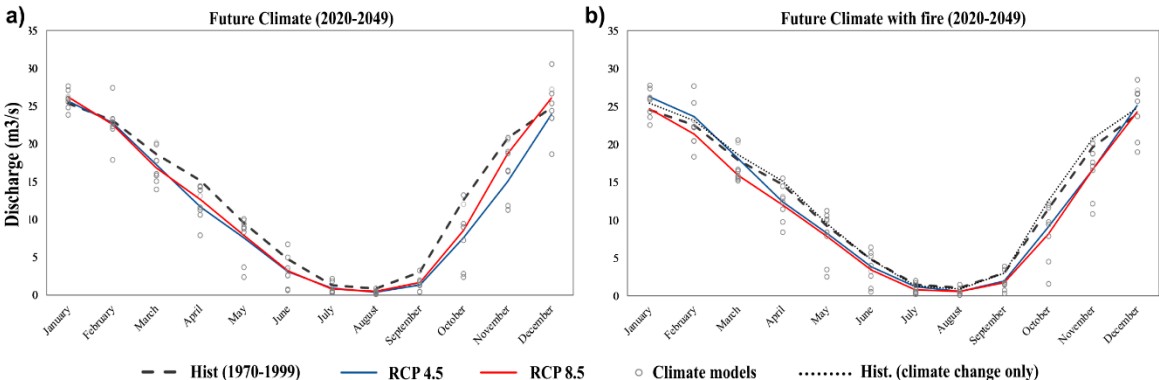

**Figure 4.** Discharge expected in the future climate (2020–2049) compared to historical period (1970–1999) (**a**) without including the effects of fire and (**b**) including the effects of fire. Circles represent values obtained for each climate model–scenario combination.

When fire is considered in model simulations, the same patterns obtained under climate change only are observed, compared to historic with fire (Figure 4b). Accordingly, a discharge reduction in spring, summer, and autumn, but a bigger increase of water in the river during winter when compared to historic with fire, especially in RCP 4.5, when more fires were modeled in SWAT (Table 3). This results in higher water in the river due to an increase of surface runoff after fire that happens in August (4–5 fires on average in 30 years). A modeling study to address the impacts of wildfire on runoff applied in a similar medium watershed in the north of Portugal observed higher runoff peaks after fire during rainy days [50].

However, a different result related to discharge when comparing both historical periods (future climate and future climate with fire) is the slight reduction of water in the river when fires are modeled (Figure 4b, comparing both historical simulations). This tendency applies to historical and RCP 8.5, but not to RCP 4.5 (Figure 4a,b). This is probably related to the occurrence (or not) of favorable meteorological conditions after fire in each scenario/climate model. Specifically, strong episodes of precipitation will lead to quick surface runoff and increase discharge, which seems to be the case of RCP 4.5 with fire. In turn, soft precipitation in combination with high temperature will foster vegetation growth, especially fast-growing herbaceous plants, shrubs, and eucalypts, with increased evapotranspiration, which seems to be the case in historical and RCP 8.5 simulations. A field-study of long-term hydrological responses to large wildfire in Australia showed that, when eucalypts regenerate, water yields are likely to decline and, conversely, when there is little or no eucalypt recovery, water yields are expected to increase [51]. It should be noted however that results are analyzed at the outlet of the watershed, where the effects on discharge are less visible when compared to the outlet of the sub-basins affected by fire.

Soil erosion is expected to slightly increase with climate change, especially under RCP 4.5, but when fires are included in SWAT simulations, soil erosion increases substantially (Figure 5a), suggesting that the effect of fires on soil erosion is more severe than the effect of climate change. Similar results were found in hydrological modeling studies with scenarios of climate change and fire applied to

a watershed in central Idaho, USA, and in Alberta, Canada, regarding sediment yield increases with fire for both historical and future climates, concluding that sediment yield is more sensitive to fire occurrence than to climate change [13,51].

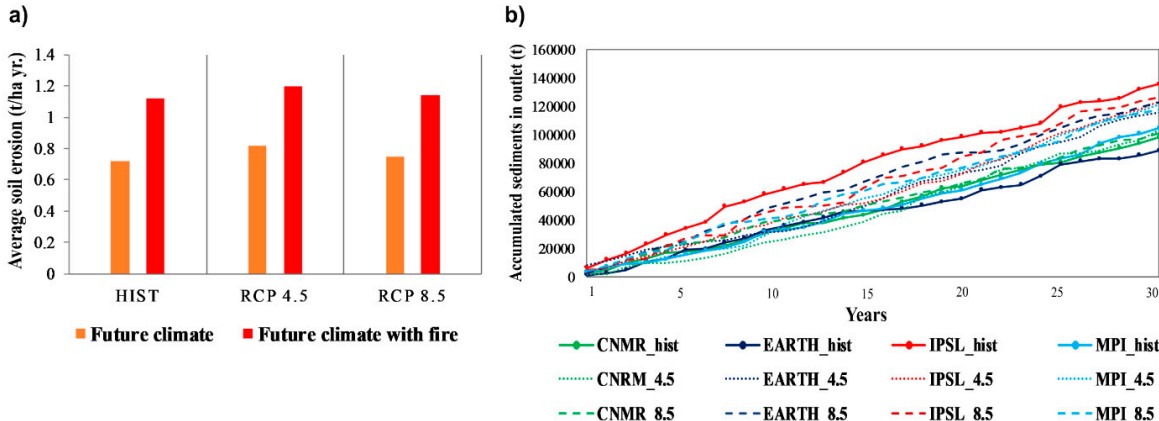

**Figure 5.** (**a**) Soil erosion under future climate (2020–2049) and future climate with fire, compared to the historical period (1970–1999). (**b**) Accumulated sediments in the river outlet in 30 years, by combination of climate model and scenario, compared with the historical period.

To have soil erosion after fire, meteorological conditions favorable to erosion, such as autumn and winter precipitation, must occur and coincide with the years with fires during the 30-year simulation. This seems to be the case of future climate under RCP 4.5, that showed a slight increase of soil erosion in the scenario of fire and climate change compared to historic, although with more fires occurring than under RCP 8.5. In general, RCP 8.5 climate scenario becomes extreme by the end of the century, and in this study, we simulate until mid-century (2049). Regarding the accumulated sediments in the river outlet throughout the 30 years of simulation, it was the historical period of CM5A-MR climate model that showed higher erosion, but the historical period of EC-EARTH that showed lower erosion (Figure 5b). Interestingly, both CM5A-MR and EC-EARTH historical simulations in SWAT considered four fires in the 30 years run period (Table 3), which seems to give more importance to post-fire meteorological conditions, namely, the coincidence of higher precipitation in winter after the fire event, than to the number of fires simulated itself. From here, climate model uncertainty can be observed and, considering that the analysis at the ensemble level (average of the four climate models, Figure 5a), must be taken with care.

### 3.4. Challenges for Landscape Adaptation to Large Fires and Future Climate

Portugal is the European country with the highest number of wildfires and the second with more burnt area, the majority of ignitions having human origin, either intentional (42%) or negligent (28%) [52]. In addition, large fires in Portugal are driven by long-standing land abandonment processes and critical weather conditions, occurring independently of large expenditures in fire-fighting resources [53]. According to our model simulations, future climate conditions are expected to increase burnt areas and, more importantly, to increase the frequency of large fire events, with hydrological consequences regarding the seasonality of river discharge and an increase of soil erosion (Figures 4 and 5). These results emphasize the need for more integrative modeling approaches to predict the combined impacts of climate change and large fires in the provision of hydrological ecosystem services and to provide sound management guidelines.

Fuel management aimed at the creation of a fire-resilient landscape, often called fire-smart management of forest landscapes, is an appealing and promising option for adaptation to climate change and novel fire regimes [54]. In fact, land abandonment in rural areas and decreasing farming activities have led to fuel accumulation, especially with fire-prone species, consequently increasing

fire hazard in the last decades [55]. Still, a survey done among European Mediterranean foresters and scientists gave more importance to adaptation measures related to firefighting efficiency and public awareness than to fuel management [56]. To better adapt rural landscapes to fire and climate change, integrated fire management, as well as communication between research and management entities, are required [56].

Traditionally, landscape adaptation management options for lowering fuel accumulation include grazing control, the creation of fire breaks, lowering tree density, and prescribed fire [56]. Interestingly, a modeling study showed an increase of 200% for future burnt areas in a scenario of "no adaptation" by 2090, but an increase below 50% in the scenario of prescribed burning (fuel control) [57]. In addition, prescribed fires can contribute to grassland recovery and the regeneration of particular plant species, and although soil proprieties can be affected, depending on initial soil characteristics, the effects are less severe than those of wildfires, because of the limited soil heating and lower fire intensity and severity [3]. In light of our results, management policies alternative to those implemented nowadays (mostly focused on fire suppression than in prevention) should aim to reduce fire severity but simultaneously ensure the long-term supply of ecosystem services and functions such as climate change mitigation or water regulation. In this sense, fire-smart strategies based upon cover type conversions from fast-growing forest species (e.g., eucalyptus plantations) to less flammable species (e.g., native oak species) have been proposed as an adaptive option to cope with climate change and increase ecosystem resistance to fire [54], with potential co-benefits for hydrological ecosystem services [27].

## 4. Conclusions

Large fires and future climate impact on soil erosion and hydrology were simulated in the Vez watershed, northwest Portugal, using the SWAT (Soil and Water Assessment Tool) hydrological model. The model was parametrized for post-fire recovery based on satellite data, which improved model replicability on vegetation dynamics after fire. To our knowledge, this was the first time SWAT was used to simulate both future climate and large fires for the mid-century period (2020–2049) in a dynamic way.

In the future (2020–2049), an increase in temperatures and a decrease in precipitation with marked uneven seasonal distribution are expected, as well as a forecasted increase in the burnt area and in the frequency of large fires, when compared to historical period (1970–1999). This will directly influence seasonal discharge in the river, with less water during spring, summer, and autumn, and more water during winter, exacerbated by large fires. Soil erosion increase is substantially higher under fire scenarios, although with a slight increase under climate change, especially under RCP 4.5 scenario that predicts more precipitation. Overall, the decrease in water supply is more influenced by climate change, whereas soil erosion increase is more dependent on fire. An important point is that meteorological conditions that follow the months after fire, which are highly random, are the critical aspect influencing discharge, but more importantly, soil erosion intensity.

The understanding of fire consequences on hydrological services provision under future climate may support decision-makers in choosing better options for landscape adaptation and post-fire mitigation, sustaining efforts to achieve SDG 7—climate action and 15—Life on Land.

**Author Contributions:** C.C.-S. conceived and designed the work, collected data, applied SWAT model for fire and climate change, and wrote the paper; B.M. prepared and analyzed LAI and NDVI data for post-fire recovery; J.P.N. ran the model for burnt areas in the future; E.P. and S.T. prepared future climate projections; A.R., A.T.M., and J.P.H. analyzed data; J.P.H. was responsible for funding acquisition; all authors supported paper writing and edits.

**Funding:** This work was carried out within the H2020 project 'ECOPOTENTIAL: Improving Future Ecosystem Benefits Through Earth Observations' (http://www.ecopotential-project.eu). The project received funding from the European Union's Horizon 2020 research and innovation programme under grant agreement No 641762. Also by Portuguese national funds through FCT—Fundação para a Ciência e a Tecnologia, I.P.,under the project FirESmart "PCIF/MOG/0083/2017". B.M. was also funded by FCT under Doctoral fellowship SFRH/BD/99469/2014, and J.P.N. under IF research grant "IF/00586/2015", through the Programa Operacional Capital Humano (POCH) co-financed by the Fundo Social Europeu and national funds from the Ministério da Ciência, Tecnologia e Ensino

Superior (MCTES). A.R. was financially supported by the Xunta de Galicia, Spain (post-doctoral fellowship ED481B2016/084-0).

**Acknowledgments:** This study was carried out under H2020 project 'ECOPOTENTIAL: Improving Future Ecosystem Benefits Through Earth Observations' (http://www.ecopotential-project.eu), grant agreement No 641762. Fire scenarios were developed under the project FirESmart "PCIF/MOG/0083/2017".

**Conflicts of Interest:** The authors declare no conflict of interest.

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
