# Peer review of "Hydrological Impacts of Large Fires and Future Climate: Modeling Approach Supported by Satellite Data"

_remotesensing, doi:10.3390/rs11232832_

Round 1

Reviewer 1 Report

This Manuscript presents an interesting and well elaborated research about the application of SWAT hydrological model and satellite data in Vez watershed in Portugal to assess the hydrological and erosional impacts of fires taking into account the future climate change. Some minor changes are suggested to further improve the manuscript.

In the section 2.2 Satellite data to support SWAT parameterization, please provide a little bit more detail about the calibration of vegetation parameters to LAI, and please provide an illustration of LAI MODIS maps used for calibration.

Also in the section 2.2 Satellite data to support SWAT parameterization, please provide a little bit more detail about the extraction of the post-fire recovery curves based on FVC and calculation of median profiles of post-fire recovery for each land cover class. Also, please provide some illustrations.

Author Response

Please see the attachment, because the the text include figures

Reviewer 2 Report

The study aims to simulate the hydrological impacts of large fires and future climate on water supply and soil erosion in Portugal. Although you can see my comments in the document itself (See attached), few important things I think the paper should focus on are:

Further discussion on how fire could lead to lower water discharge, and impact of fire under climate change.  Further clarity on differences in soil erosion  under future climate and future climate with fire scenarios.  Can the estimates of fire frequency/burnt areas under climate change be improved with models like random forests, neural networks compared to linear models.

Author Response

::::::::::::::::::::::::::::::::::::::::::::::::::::::::::::::::::::::::::::::::::::::::::::::::::::::::::::::::::::::::::::::::::::::::::::::::::::::::::::::

Reviewer 2

::::::::::::::::::::::::::::::::::::::::::::::::::::::::::::::::::::::::::::::::::::::::::::::::::::::::::::::::::::::::::::::::::::::::::::::::::::::::::::::

The study aims to simulate the hydrological impacts of large fires and future climate on water supply and soil erosion in Portugal. Although you can see my comments in the document itself (See attached), few important things I think the paper should focus on are:

R1: We thank the Reviewer for the positive remarks and encouraging comments. The changes suggested in the pdf provided were included in the manuscript (highlighted in yellow).

Further discussion on how fire could lead to lower water discharge, and impact of fire under climate change. 

R2: R2: We thank the Reviewer for the remark. Here, we have two opposite situations, but with support arguments for both. When comparing future discharge with the respective historical records, fire should exacerbate the tendency observed under climate change only, especially during winter with more discharge in the river, but the differences are slight. It should be noted however that results are analysed at the outlet of the watershed, where the effects on discharge are less visible. But when comparing both historical periods (with and without fire), there is a reduction in discharge, resulted from a slight increase in evapotranspiration. This applies to historical and RCP 8.5 (that in this case is not so extreme as RCP 4.5, because we only modelled until mid-century), but not to RCP 4.5. We believe this is related to the combination of favourable meteorological conditions after fire in each scenario/climate model. Specifically, strong episodes of precipitation will lead to quick surface runoff and increased discharge, which seems to be the case of RCP 4.5 with fire. In turn, soft precipitation in combination with high temperature will foster vegetation growth and increased evapotranspiration, which seems to be the case in historical and RCP 8.5 simulations. What can be concluded from here is that meteorological conditions that follow the months after fire, which are highly unpredictable, are the trigging aspect influencing discharge.

In line with the above, new text was added in the manuscript. Please see lines: 404-418: “However, a different result related to discharge when comparing both historical periods (future climate and future climate with fire), is the slight reduction of water in the river when fires are modelled (Figure 4b comparing both historical simulations). This tendency applies to historical and RCP 8.5, but not to RCP 4.5 (Figure 4 a/b). This is probably related to the occurrence (or not) of favourable meteorological conditions after fire in each scenario/climate model. Specifically, strong episodes of precipitation will lead to quick surface runoff and increase discharge, which seems to be the case of RCP 4.5 with fire. In turn, soft precipitation in combination of high temperature will foster vegetation growth, especially fast-growing herbaceous plants, shrubs and eucalypts, with increased evapotranspiration, which seems to be the case in historical and RCP 8.5 simulations. A field-study of long-term hydrological responses to large wildfire in Australia showed that, when eucalypts regenerate, water yields are likely to decline and, conversely, when there is little or no eucalypt recovery, water yields are expected to increase [51]. It should be noted however that results are analysed at the outlet of the watershed, where the effects on discharge are less visible when compared to the outlet of the subbasins affected by fire.

And also, in the conclusion section. Please see lines: 501-503: “An important point is that meteorological conditions that follow the months after fire, which are highly random, are the trigging aspect affecting discharge, but more importantly soil erosion intensity.”

Further clarity on differences in soil erosion under future climate and future climate with fire scenarios. 

R3: We thank the Reviewer for the remark. In the pdf the following related comment was raised: it seems like there is no increase in soil erosion with fire under extreme climate change scenario i.e., RCP 8.5. In fact, RCP 8.5 only becomes extreme by the end of the century, when the concentration of emissions is higher than in RCP 4.5.  Here we carry simulations until mid-century (2049), therefore results should be interpreted without assuming that RCP 8.5 is worse than RCP 4.5. It depends on the meteorological conditions that follow after fire, which are unpredictable. In previous studies, we obtained the same results. The following sentence was added to the manuscript to clarify this point. Please see lines 436-437: “In general, RCP 8.5 climate scenario becomes extreme by the end of the century, and in this study, we simulate until mid-century (2049).”

Can the estimates of fire frequency/burnt areas under climate change be improved with models like random forests, neural networks compared to linear models.

R4: We acknowledge that more sophisticated statistical approaches could be used to improve the robustness of burned area predictions; we have now mentioned this in the results and discussion by adding three examples where such approaches (non-linear regressions and random forests) have been used for this purpose. Please see lines 368-371: “In the future, more sophisticated burned area models using more variables and non-linear relationships [45,46] or more sophisticated approaches such as random forests and boosted regression trees [47] can be used to improve the robustness of burned area predictions.” The new references are:

Barbero R., Abatzoglou J. T., Larkin N. K., Kolden C. A., Stocks B. (2015) Climate change presents increased potential for very large fires in the contiguous United States. International Journal of Wildland Fire 24, 892-899. https://doi.org/10.1071/WF15083

Mann M. L., Batllori E., Moritz M. A., Waller E. K., Berck P., Flint A. L., Flint L. E., Dolfi, E. (2016). Incorporating Anthropogenic Influences into Fire Probability Models: Effects of Human Activity and Climate Change on Fire Activity in California. PLOS ONE, 11(4), e0153589. doi:10.1371/journal.pone.0153589

Liu, Z., & Wimberly, M. C. (2016). Direct and indirect effects of climate change on projected future fire regimes in the western United States. Science of The Total Environment, 542, 65–75. doi:10.1016/j.scitotenv.2015.10.093

Reviewer 3 Report

The paper entitled "Hydrological impacts of large fires and future climate: modelling approach supported by satellite data" is a timely and relevant paper that would be of interest to a wide readership. The manuscript directly  speaks to the need to simulate future scenarios related to climate change and induced increased fire severity for source water protection and water supply. The paper is well written and informative. The conclusions are reasonable. 

Considerations for minor changes

pg ln 58 The first line of the paper speaks to the need to the threat of severe wildfire on soil erosion and water supply. I suggest adding reference to the manuscript, that speak more directly to this issue.

Emelko et al, 2011. Implications of land disturbance on drinking water treatability in a changing climate. Wat. Res. 45:461.

pg 12 ln 412. The authors point to a study that concluded "sediment yield is more sensitive to fire occurrence than to climate change". Given the wide readership it would be useful to provide additional supporting literature as the citation below that provides measured data illustrating this phenomenon.

Silins et al, 2009. Sediment production following severe wildfire and post-fire salvage logging in Rocky Mountain headwaters. Catena, 38:3:189.

Author Response

::::::::::::::::::::::::::::::::::::::::::::::::::::::::::::::::::::::::::::::::::::::::::::::::::::::::::::::::::::::::::::::::::::::::::::::::::::::::::::::

Reviewer 3

::::::::::::::::::::::::::::::::::::::::::::::::::::::::::::::::::::::::::::::::::::::::::::::::::::::::::::::::::::::::::::::::::::::::::::::::::::::::::::::

The paper entitled "Hydrological impacts of large fires and future climate: modelling approach supported by satellite data" is a timely and relevant paper that would be of interest to a wide readership. The manuscript directly  speaks to the need to simulate future scenarios related to climate change and induced increased fire severity for source water protection and water supply. The paper is well written and informative. The conclusions are reasonable.

Considerations for minor changes

pg ln 58 The first line of the paper speaks to the need to the threat of severe wildfire on soil erosion and water supply. I suggest adding reference to the manuscript, that speak more directly to this issue.

Emelko et al, 2011. Implications of land disturbance on drinking water treatability in a changing climate. Wat. Res. 45:461.

pg 12 ln 412. The authors point to a study that concluded "sediment yield is more sensitive to fire occurrence than to climate change". Given the wide readership it would be useful to provide additional supporting literature as the citation below that provides measured data illustrating this phenomenon.

Silins et al, 2009. Sediment production following severe wildfire and post-fire salvage logging in Rocky Mountain headwaters. Catena, 38:3:189.

R1: We appreciate the Reviewer’s positive comments and thank him/her for suggesting new references that improved the readability of the sentences. The new references were included and highlighted in yellow in the manuscript.

Round 2

Reviewer 2 Report

Thank you for addressing the comments.